# High prevalence of long-term olfactory disorders in healthcare workers after COVID-19: A case-control study

**Johannes Frasnelli[1,2☉], Arnaud Tognetti[1,3☉], Anja L. Winter[1], Evelina Thunell[1], Mats J. Olsson[1], Nina Greilert[4], Jonas K. Olofsson[5], Sebastian Havervall[4], Charlotte Thålin[4‡], Johan N. Lundström[1,6‡]***

1 Department of Clinical Neuroscience, Karolinska Institutet, Stockholm, Sweden, 2 Department of Anatomy, Université du Québec à Trois-Rivières, Trois-Rivieres, QC, Canada, 3 CEE-M, CNRS, INRAE, Institut Agro, University of Montpellier, Montpellier, France, 4 Department of Clinical Sciences, Karolinska Institutet, Danderyd Hospital, Stockholm, Sweden, 5 Department of Psychology, Stockholm University, Stockholm, Sweden, 6 Monell Chemical Senses Center, Philadelphia, PA, United States of America

☉ These authors contributed equally to this work.
‡ CT and JNL are Joint Senior Authors
* Johan.Lundstrom@ki.se

**Data Availability Statement:** The datasets generated along with associated scripts and figures are available on the Open Science Framework data

## Abstract

### Background

More than a year after recovering from COVID-19, a large proportion of individuals, many of whom work in the healthcare sector, still report olfactory dysfunctions. However, olfactory dysfunction was common already before the COVID-19 pandemic, making it necessary to also consider the existing baseline prevalence of olfactory dysfunction. To establish the adjusted prevalence of COVID-19 related olfactory dysfunction, we assessed smell function in healthcare workers who had contracted COVID-19 during the first wave of the pandemic using psychophysical testing.

### Methods

Participants were continuously tested for SARS-CoV-2 IgG antibodies since the beginning of the pandemic. To assess the baseline rate of olfactory dysfunction in the population and to control for the possibility of skewed recruitment of individuals with prior olfactory dysfunction, consistent SARS-CoV-2 IgG naïve individuals were tested as a control group.

### Results

Fifteen months after contracting COVID-19, 37% of healthcare workers demonstrated a quantitative reduction in their sense of smell, compared to only 20% of the individuals in the control group. Fifty-one percent of COVID-19-recovered individuals reported qualitative symptoms, compared to only 5% in the control group. In a follow-up study 2.6 years after COVID-19 diagnosis, 24% of all tested recovered individuals still experienced parosmia.

depository https://osf.io/hja2p/?view_only=
53d8bb21c06c48d2bee231f75797789c.

**Funding:** Funding was provided by grants awarded
to JNL and CT from the Knut and Alice Wallenberg
Foundation (KAW 2018.0152 and KAW 2020.0182,
respectively), the Swedish Research Council
(2021-06527) and a donation from Stiftelsen Bygg-
Göta för Vetenskaplig forskning to JNL, as well as
the Swedish Heart Lung Foundation to CT. JF is
supported by the Fonds de Recherche du Québec –
Santé, the Natural Sciences and Engineering
Council of Canada (RGPIN-2022-04813) and the
Canadian Institutes of Health Research (PJT
173514). No funder took any part in planing,
execution, analyses, or reporting of this research.

**Competing interests:** The authors have declared
that no competing interests exist.

## Conclusions

In summary, 65% of healthcare workers experienced parosmia/hyposmia 15 months after
contracting COVID-19. When compared to a control group, the prevalence of olfactory dys-
function in the population increased by 41 percentage points. Parosmia symptoms were still
lingering two-and-a half years later in 24% of SARS-CoV-2 infected individuals. Given the
amount of time between infection and testing, it is possible that the olfactory problems may
not be fully reversible in a plurality of individuals.

## Introduction

Olfactory dysfunction is the most specific symptom of acute COVID-19 [1–3] and our ability
to smell can be heavily impacted by the disease [4]. In many patients, olfactory function is
regained after the acute phase [5], yet a non-negligeable proportion of patients exhibits chronic
dysfunction [6]. Estimations of the prevalence of olfactory dysfunction 6 months after
COVID-19, the threshold for chronicity of olfactory dysfunction, vary widely [5,7–13]. A
recent assessment of olfactory dysfunction based on subjective reports suggests that a stagger-
ing 61% still experience olfactory dysfunction two years after infection [6].

Despite the clear evidence that COVID-19 can result in long-term olfactory dysfunction,
there are several caveats in current prevalence estimates that may account for the large differ-
ences in reported estimates. First, most of the estimations are based on self-reported olfactory
dysfunction which is notoriously unreliable [14,15]. Second, COVID-19 status is seldom deter-
mined by biological assays but rather by self-reported status, which can bias the estimations.
Third, the prevalence of olfactory dysfunction unrelated to COVID-19 is usually not consid-
ered. In fact, olfactory dysfunction in the general population was estimated to be around 20%
before the pandemic [16–19], with leading causes being sinonasal diseases, traumatic brain
injuries, and viral infections of the upper respiratory tract [20]. Thus, an accurate estimate of
the prevalence of olfactory dysfunction after COVID-19 infection requires consideration of
the prevalence of olfactory dysfunction in a sample where COVID-19 infection can be ruled
out.

Olfactory dysfunction encompasses various impairments to the sense of smell, categorized
into quantitative dysfunction, involving a reduction or loss of olfactory function (hyposmia or
anosmia) [19], and qualitative dysfunction, which involves altered odor perception such as
parosmia or phantosmia. Standardized assessments of parosmia and phantosmia can currently
only be carried out using questionnaires [21] due to their subjective nature. While self-evalua-
tion questionnaires are commonly used to assess quantitative dysfunction, their precision war-
rants caution in interpretation [14,15]. Therefore, when estimating the prevalence of olfactory
dysfunction after COVID-19, a comprehensive approach should incorporate both question-
naires and well-validated gold standard psychophysical tests [22,23] for a more accurate assess-
ment of objective and subjective aspects.

Healthcare workers may be particularly affected by long-term consequences of COVID-19.
For example, mental health disorders were particularly high in this population both during
and following COVID-19 [24,25]. In line with this, nearly half of infected healthcare workers
reported reduced chemosensory abilities one year after COVID-19 [5], which is considerably
higher than in the general population [26]. It is therefore particularly valuable to study the
long-term effects of COVID-19 on olfactory function in this group. We designed this study to

assess olfactory dysfunction in a sample of healthcare workers where continuous serum assessments of SARS-CoV-2 infection were available. Specifically, we included two groups of participants; one group that had undergone SARS-CoV-2 infection and a control group that had not, as confirmed by the continuous serum testing. Our objective was to establish the prevalence of olfactory dysfunction due to COVID-19 up to 15 months after infection by evaluating olfactory dysfunction with validated olfactory tests in both groups. We hypothesized that the group who had undergone SARS-CoV-2 infection would be afflicted by a significantly higher prevalence of olfactory dysfunction compared to the control group.

## Materials and method

### Participants

All participants were recruited from the ongoing COMMUNITY (COVID-19 Immunity) study that enrolled 2149 healthcare workers employed at Danderyd Hospital, Stockholm in April 2020. All participants were tested for SARS-CoV-2 IgG antibodies every four months since the beginning of the pandemic [2], see https://ki.se/en/kids/community for more information. We screened out individuals with acute nasal congestion/rhinorrhea, neurodegenerative diseases, and other conditions associated with reduced olfactory function. All participants gave written informed consent, and the study was approved by the Swedish Ethical Review Authority (dnr 2021–02052).

We recruited healthcare workers that had contracted COVID-19 during the first wave in Stockholm between January and May 2020. Of a total of 320 SARS-CoV-2 IgG positive individuals, 100 participated in the current study. Two were excluded due to an inability to perform the tests, leaving a final sample of 98 COVID+ participants (average age: 48 years; SD ±12; 84 women; average time since infection 447, SD ±73, days). All participants had experienced mild COVID-19 and had not required hospitalization. We further invited an equal number of SARS-CoV-2 naïve, meaning SARS-CoV-2 IgG negative during all sampling timepoints since enrollment, healthcare workers from the same cohort with a similar age profile. A total of 44 individuals were included, with 3 individuals subsequently excluded due to olfactory-related sickness, excessive construction noise during testing precluding concentration, and refusal to perform some tests. This resulted in a final control sample of 41 COVID- participants (average age: 51 years; SD ±11; 38 women).

### Chemosensory assessments

**Quantitative olfactory dysfunction.** Quantitative olfactory dysfunction was assessed using two different approaches. First, we assessed subjective quantitative olfactory function using a 10-point visual analog scale. Participants responded to the question "*How has your sense of smell been during the last three days*?"; the scale ranged from 0 (no smell) to 10 (very good sense of smell). Second, we assessed objective quantitative olfactory function psychophysically, using the Sniffin' Sticks test battery [23]. This test is based on felt-tip pen-like odor dispensing devices and allows for separate assessments of the ability to discriminate (D) and identify (I) odors as well as an olfactory detection threshold (T). To assess odor quality discrimination, 16 triplets of pens were presented to the participant. Each triplet consisted of two pens with identical odorants and one with an odorant of different quality. To evaluate odor identification abilities, we used a forced-choice cued identification task using 16 different odorants. Each odor was presented together with a cue card listing four alternative odor labels, and the participant picked the label that best described the quality of the perceived odor. To estimate odor detection thresholds, we used the odor n-Butanol in a three-alternative forced-choice staircase procedure with seven reversals in a 16-step binary dilution series. The

individual sub-scores were then combined to a global TDI score, for which normative data are available and allows for the diagnosis of normosmia, hyposmia, and functional anosmia [27].

**Qualitative olfactory dysfunction.** Qualitative olfactory dysfunction was assessed using a questionnaire containing four items [21]. The questionnaire addressed aspects of qualitative olfactory dysfunction with regards to alterations in food perception, the presence of odors in absence of an odor source, pleasantness of perceived odors, and the impact of the perception of altered odors; participants could respond using a 4-point scale ranging from 1 (never) to 4 (always). We analyzed the questionnaire in two ways: first, we calculated the sum score of all four items (Parosmia score). Then, we counted the number of participants who responded *always*, *often*, or *rarely* to the question "*the biggest problem is not that I do not or weakly perceive odors, but that they smell different than they should* vs those who responded *never* (Parosmia presence).

## Serological analyses of antibodies

A detailed description of serological analyses has been presented elsewhere [2]. Briefly, IgG reactivity was measured towards three different SARS-CoV-2 virus protein variants, Spike trimers, Spike S1 domain, and Nucleocapsid protein, and analyzed using a multiplex antigen bead array in high throughput 384-plates form at using a FlexMap3D (Luminex Corp) [28]. To be assigned to the SARS-CoV-2 IgG positive group, reactivity against at least two of the three different variants of the viral antigens was required, calculated to have 99.2% sensitivity and 99.8% specificity [2]. We did not determine individual virus variants but random sampling of the Stockholm population during the time participants in this study were infected showed that three main strains of Variants Being Monitored (VBM) of the SARS-CoV-2 dominated, namely the Wildtype, and to a lesser extent, the B.1.1/B.1.1.29 and B.1.1.1/C.14 [29].

## Procedure

Olfactory dysfunction data were collected between June and November 2021. Upon inclusion, participants responded to a questionnaire containing the VAS and the parosmia questions before the Sniffin' Sticks test battery was administered. The total testing time was 1.5 h and participants received monetary compensation for their participation.

## Follow-up

To further estimate the persistence of qualitative olfactory dysfunction, participants diagnosed with established parosmia were contacted approximately one year after their initial participation for a follow-up conducted via phone. Out of the 48 participants contacted, 41 individuals responded to the follow-up assessment (with an average time since infection of 963 days, i.e., over 2.6 years since onset; SD ±64 days), which featured identical questions and response options as those used during the initial assessment.

## Statistics

To assess whether an infection with SARS-CoV-2 correlated with long-term quantitative and/ or qualitative olfactory dysfunction we compared scores on the (a) parosmia questionnaire (4 COVID+ and 3 COVID- individuals refrained from answering this questionnaire), (b) quantitative subjective olfactory dysfunction scale, and (c) Sniffin' Sticks test between COVID+ and COVID- groups. To correct for the uneven sample size between the two groups as well as the non-normal distribution of the variables, we performed non-parametric statistical group comparisons using a two-sided Wilcoxon rank sum test with continuity correction. We also compared frequencies between groups using chi-square tests.

## Results

First, we assessed quantitative olfactory dysfunction. To do so, we explored potential differences in subjective performance: here, participants from the COVID+ group (N = 98) evaluated their olfactory function as significantly worse, 6.9 (*SD* ±2.5) out of a total of 10 points compared to the COVID- group (N = 41), 8.9 points (*SD* ±1.2; *W* = 955; N = 139, *p* < .0001). In line with the subjective experience, the average Sniffin' Sticks TDI score was also significantly lower in the COVID+ group according to a Wilcoxon rank sum test, 30.9 points (*SD* ±5.9), compared to the COVID- group, 34.0 points (*SD* ±3.4; *W* = 1416.5; N = 139, *p* = .006; Fig 1A). Interestingly, upon separate analysis of the TDI subscales, only discrimination (*W* = 1293; N = 139, p = .0008; Fig 1C) and identification (*W* = 1213.5; N = 139, p = .0002; Fig 1D) subscores were significantly lower in the COVID+ group (D = 11.66 ± 2.5, I = 12.11 ± 2.5) compared to the COVID- group (D = 13.19 ± 1.9, I = 13.68 ± 1.3). However, no statistically significant difference (*W* = 2107; N = 139, p = .65; Fig 1B) was observed for the threshold subscore between the two groups (COVID+ = 7.10 ± 2.4, COVID- = 7.11 ± 2.0). We then assessed the proportion of individuals with a clinically relevant quantitative olfactory dysfunction. Based on the TDI score, the frequency of quantitative olfactory dysfunction was significantly higher ($X^2$ (1, N = 139) = 6.28, *p* = .01) in the COVID+ group compared to the COVID- group. A total of 37% of the COVID+ group suffered quantitative olfactory dysfunction, with 4 individuals exhibiting anosmia (TDI score = 14.88 ± 2.0) and 32 individuals exhibiting hyposmia (TDI score = 26.11 ± 3.9). In the COVID- group, 20% (N = 8) showed quantitative olfactory dysfunction, all of whom had hyposmia (TDI score = 29.56 ± 0.8). The average TDI scores for individuals with normosmia were 34.38 (SD = 2.6) and 35.06 (SD = 2.8) for the COVID+ and the COVID- groups, respectively. Last, we investigated whether the degree of quantitative olfactory dysfunction was influenced by the time since COVID-19 infection and

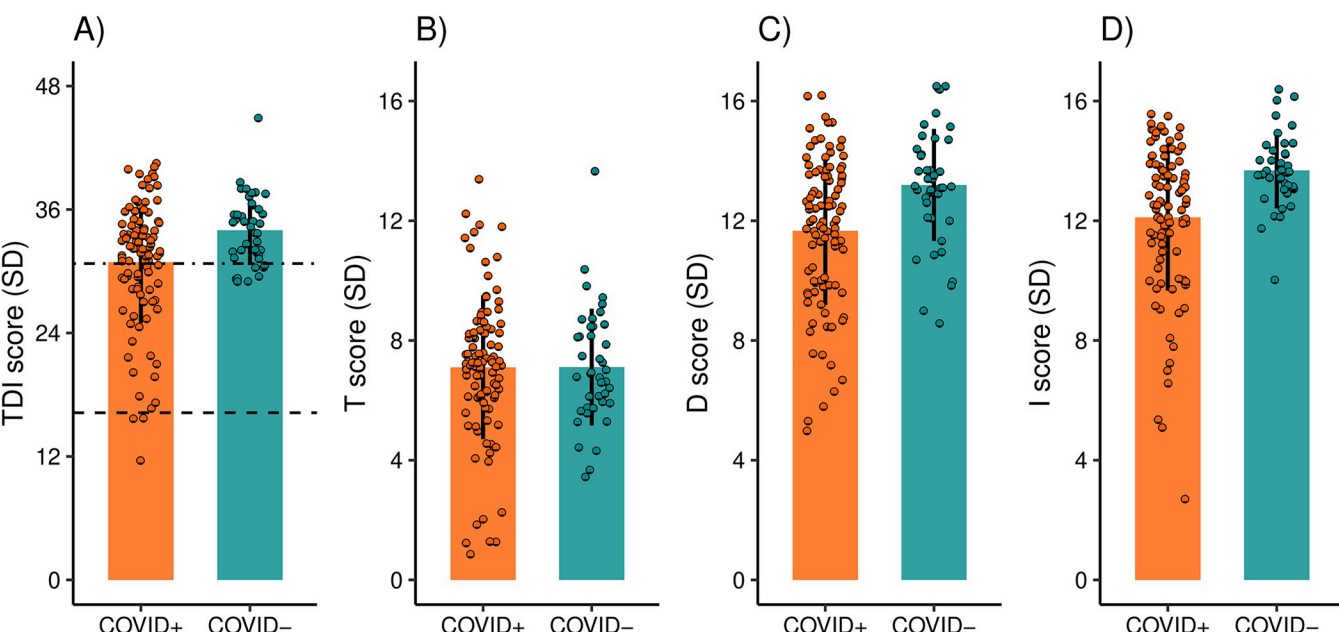

**Fig 1. Quantitative olfactory dysfunctions for COVID-19 positive (COVID+, N = 98) and COVID-19 naïve (COVID-, N = 41) groups.** (A) Mean TDI scores for COVID+ and COVID- groups with error bars denoting standard deviation (SD) as well as values for each participant indicated by circles. Dashed lines indicate cut-off scores for the clinical diagnoses Hyposmia (threshold score: 30.75) and Anosmia (threshold score: 16.25). Higher TDI scores (possible range 1–48) indicate better olfactory performance. TDI subscale scores for the COVID+ and COVID- groups are presented with error bars denoting standard deviation (SD) for (B) Threshold, (C) Discrimination, and (D) Identification scores separately.

individuals' age. A Spearman correlation test indicated that the TDI score of individuals from the COVID+ group was not significantly affected by the number of days since infection (N = 71, $r$ = -.1, $p$ = .43; see S1A Fig). However, TDI significantly declined with age in the COVID+ group (Spearman correlation test, $r$ = -.28, $p$ = .005, S1B Fig). It's worth noting that in the COVID- group, a similar negative relationship between TDI and age was observed, although it did not reach statistical significance ($r$ = -.27, $p$ = .09, S1B Fig).

Next, we assessed to what level individuals in the respective groups experienced qualitative olfactory dysfunction using the 4-item parosmia scale. The average score was significantly higher in the COVID+ group, 2.8 points (*SD* ±2.8), compared to the COVID- group, 0.7 points (*SD* ±1.2; Wilcoxon rank sum, $W$ = 2595; N = 132, $p$ < .0001; Fig 2A). Correspondingly, we found that significantly more individuals from the COVID+ group reported parosmia (48 of 94 participants, 51%) relative to individuals from the COVID- group (2 of 38 participants, 5%; $X^2$ (1, N = 132) = 22.2, $p$ < .0001). To understand the degree of symptom severity, we also assessed the answers to the parosmia severity question in each group. As can be seen in Fig 2B, slightly more than half of the individuals in the COVID+ group who reported parosmia symptoms experienced only minor symptoms (None 49%, Mild, 28%, Medium 16%, Severe 7%). This means that about a quarter of all individuals in the COVID+ group experienced medium to severe parosmia symptoms 15 months after COVID-19.

Further, we followed up with 48 individuals from the COVID+ group with established parosmia 2.6 years after COVID-19 infection. Of the 41 individuals who responded, 23 (56% of sub-sample, 24% of total initial COVID+ sample) still experienced parosmia. In fact, 41% indicated that they experienced medium to severe parosmia symptoms (None 43.9%, Mild, 14.6%, Medium 31.7%, Severe 9.8%). This long-lasting qualitative olfactory dysfunction was also reflected by a high parosmia score (3.1 ± 2.3 points).

As to be expected, there was a considerable comorbidity between diagnoses. While 35% of the COVID+ group (total N = 94) experienced no olfactory dysfunction at all, 14% experienced anosmia/hyposmia but no parosmia, 29% experienced parosmia but no anosmia/hyposmia, and 22% experienced both anosmia/hyposmia and parosmia. In other words, a total of 65% of COVID+ participants experienced some form of olfactory dysfunction on average 15

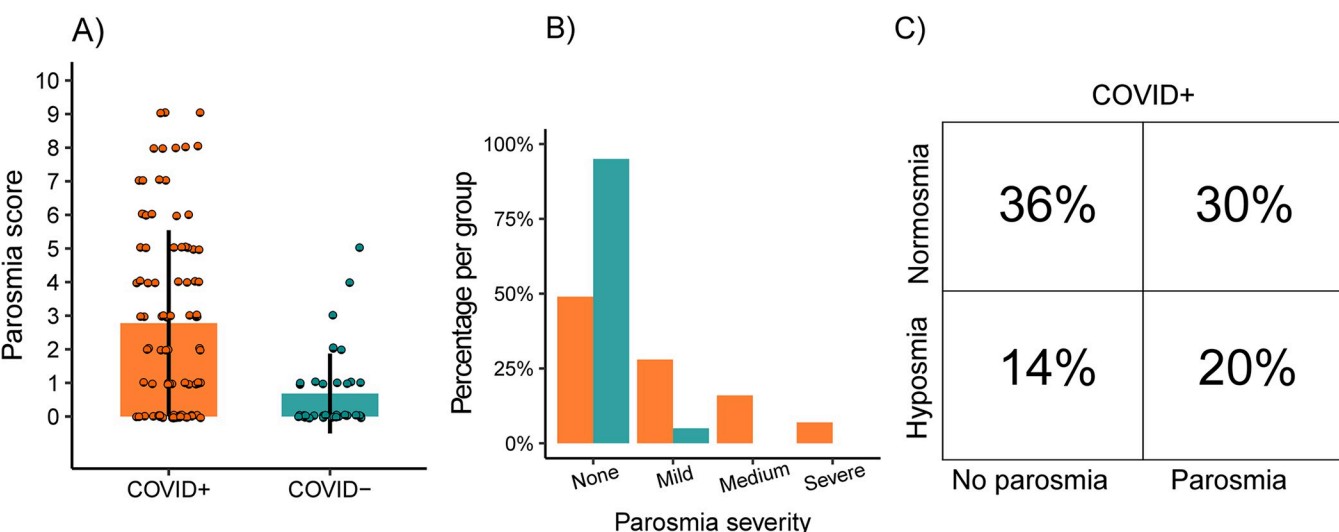

**Fig 2. Distribution of parosmia score.** A) Distribution of scores per patient group (N = 94), COVID-19 positive (COVID+) and COVID-19 naïve (COVID-, N = 38), on the Landis parosmia questionnaire. B) Percentage of individuals in each patient group, grouped by reported parosmia severity score. C) Proportion of COVID+ participants (N = 94) in each olfactory dysfunction classification group.

months after COVID-19 (Fig 2C). This frequency was lower in the COVID- group (N = 38) where only 24% experienced some form of olfactory dysfunction, 76% experienced no olfactory dysfunction, none had anosmia, 18% experienced hyposmia but no parosmia, 3% experienced parosmia but no hyposmia, and 3% experienced both hyposmia and parosmia). As expected, the frequency of combined quantitative and qualitative olfactory dysfunction was significantly lower in the COVID- group compared to the COVID+ group ($X^2$ (1, N = 132) = 32.4, $p < .0001$). This means that COVID-19 increased olfactory dysfunction (parosmia, hyposmia, or parosmia and hyposmia) with 41 percentage points (COVID+: 65%; COVID-: 24%) 15 months after COVID-19 onset.

## Discussion

More than a year after recovering from COVID-19, nearly two thirds (65%) of participating healthcare workers still exhibited some form of olfactory dysfunction with more than a third (37%) showing a clinically reduced sense of smell. In comparison, approximately a quarter (24%) of individuals without prior SARS-CoV-2 infection displayed clinically altered sense of smell. Further, amongst the individuals who experienced SARS-CoV-2 infection, about half (51%) experienced parosmia, compared to only 5% in the SARS-CoV-2 naive group; an increase of parosmia due to COVID-19 with 46 percentage points. More specifically, the first wave of COVID-19 nearly doubled the prevalence of olfactory dysfunction in this population from 20% to 36% and increased the prevalence of any form of olfactory dysfunction by a full 41 percentage points compared to individuals whose immune system was never exposed to the virus. Perhaps even more striking, 24% of all tested COVID-19 survivors still experience parosmia 2.6 years after COVID-19 diagnosis, nearly half of which experience medium to severe symptoms.

Olfactory dysfunction is common also in the general population. Before the COVID-19 pandemic, the rate of quantitative olfactory dysfunction was consistently estimated to approximately 20% [16,19]. In our control sample of COVID-19 naïve individuals, we found a similar percentage with 20% of the participants exhibiting quantitative olfactory dysfunction. As outlined above, olfactory dysfunction can have different etiologies, including sinonasal disease, traumatic brain injury, neurodegenerative diseases, and more [20,30]. It is therefore crucial to assess the prevalence of olfactory dysfunction due to COVID-19 against this background of olfactory dysfunction in the population. Our data suggests that COVID-19 roughly doubles the prevalence of quantitative olfactory dysfunction in the general population. A different and bleaker picture emerges when considering also qualitative olfactory dysfunction, i.e., parosmia and phantosmia. These conditions are relatively rare in the general population, which is also reflected by the prevalence of 5% in the COVID-19 naïve group aligning with prior assessments of Swedish samples [31,32]. However, nearly half of the COVID+ group exhibited qualitative olfactory dysfunction on average 15 months after contracting the disease.

Our study shows that 37% of COVID+ individuals still experienced reduction in their olfactory performance, even on average 15 months after COVID-19 onset. This percentage is higher compared to similar studies, see among others [13]. Notably, while there was a significant difference in overall olfactory functions between the two groups, as operationalized by the TDI scores, this difference was restricted to performance on the odor quality discrimination and odor identification subtests, but not in odor detection threshold This finding suggests that a potential underlying driver of the differences is the large difference in parosmia that might distort the odor quality of the odors included in the two-odor subtest.

It is not yet completely understood how a SARS-CoV-2 infection leads to olfactory dysfunction. The leading explanation of the acute olfactory dysfunction seen in patients is linked to

the ability of the virus to infect human cells that co-express ACE2 and TMPRSS2 proteins, such as the sustentacular cells of the olfactory mucosa [33]. Upon infection, these cells degenerate, which disturbs the local environment and crucially results in cell death of olfactory receptor neurons and consequently olfactory dysfunction [33,34]. The olfactory system demonstrates however a very good ability to regenerate [35], which might explain why most SARS-CoV-2 infected individuals regain olfactory abilities within weeks following the acute phase [5,9]. It is not yet clear why some individuals do not completely regain their olfactory abilities.

Recent data from the verbal track-and-trace program in the United Kingdom suggests that fewer individuals report subjective olfactory dysfunction after infection with the later Omicron (BA.1. and BA.2) variants than the original virus variants [36]. We do not know what specific virus variants individuals in our sample was infected with or exactly what proportions of virus variants dominated in our specific sample, but random sampling of the Stockholm population at the time indicated that the Wildtype and, to a lesser extent the B.1.1/B.1.1.29 and B.1.1.1/C.14 strains, dominated [29]. Although tentative data suggest that fewer individuals report subjective olfactory dysfunction after Omicron variant infection, these are based on subjective data collected only a day or two after testing positive. Whether potential lower numbers of olfactory dysfunction after Omicron or other later variants could be due to a delay in onset of olfactory dysfunction that in turn might affect long-term outcomes, remains to be determined. Future studies should address differences between virus variants in effects on olfactory function.

A significant strength of the present study is that all participants were continuously serologically monitored from the onset of the pandemic, meaning that it can be firmly established not only that all participants in the COVID+ group had undergone a SARS-CoV-2 infection, but also when. Critically, we can firmly claim that no participants in the COVID- group, a control group from the same cohort, had seroconverted at any point before sensory testing, thereby giving us a true baseline for existing olfactory dysfunction prevalence. Likewise, most studies assessing olfactory dysfunction in a general population suffer from collider bias, i.e., that individuals that experience olfactory problems are more likely to volunteer for the study in the first place, thereby erroneously increasing the prevalence of dysfunction. By using a control group undergoing the same recruitment strategy as the target group and the population being healthcare professionals, possibly better informed and willing to participate in research without personal gain, it can be assumed that collider bias acts in equal strength across groups. It can be argued that healthcare workers, in general, are a group consisting of generally more healthy individuals than the general population. Olfactory problems and their underlying mechanisms therefore could be different than other populations. However, the fact that we obtained identical prevalence of olfactory dysfunction in the COVID- group as the only two previously published prevalence studies in a Swedish sample do not support this notion [31,32]. That said, the strengths mentioned come from the fact that we sampled from a smaller group of closely monitored individuals with a profession that was unfortunately highly taxed for time during a pandemic. This meant that obtaining data from a large and diverse sample was not possible. In addition to the relatively small sample sizes, the COVID+ group had nearly double the participants compared to the COVID- groups, which we aim to statistically control for by using methods such as the Wilcoxon rank sum test. These tests do not assume equal sample sizes and are therefore robust against differences in sample size. Another weakness is that the sex ratio in both groups were skewed with more women participating, which, however, also reflects the underlying population of healthcare workers. Women are also known to be slightly more prone to experiencing long-term effects on olfactory function after upper respiratory

infections [37]. Whether the skewed sex ratio of our sample affects the generalization of our results to other populations of healthcare workers is not known.

## Conclusion

We show that COVID-19 nearly doubles the already large prevalence of quantitative olfactory dysfunction to approximately 37% in a sample of healthcare workers. Furthermore, about half of the COVID-19 survivors exhibit qualitative olfactory dysfunction. Finally, nearly two thirds (65%) of COVID-19 survivors in this group of healthcare workers exhibit olfactory dysfunction of some form 15 months and 24% reported parosmia 2.6 years after infection. Given the length of time, it is possible that these olfactory problems may not be fully reversible in a plurality of individuals.

## Supporting information

**S1 Fig.** Scatterplot illustrating the association between TDI score and (A) days since infection or (B) age of individuals who had had COVID-19 (COVID+, in orange) and COVID-19 naïve individuals (COVID-, in blue). Spearman correlations showed that TDI score was not significantly affected by the number of days since infection (N = 71, $r$ = -.1, $p$ = .43), but significantly declined with age in the COVID+ group ($r$ = -.28, p = .005) but not in the COVID- group ($r$ = -.27, $p$ = .09).
(PDF)

## Author Contributions

**Conceptualization:** Jonas K. Olofsson, Sebastian Havervall, Charlotte Thålin, Johan N. Lundström.

**Data curation:** Arnaud Tognetti, Anja L. Winter, Mats J. Olsson, Nina Greilert, Charlotte Thålin.

**Formal analysis:** Johannes Frasnelli, Arnaud Tognetti, Anja L. Winter, Evelina Thunell, Johan N. Lundström.

**Funding acquisition:** Johannes Frasnelli, Charlotte Thålin, Johan N. Lundström.

**Investigation:** Anja L. Winter, Nina Greilert.

**Methodology:** Sebastian Havervall, Johan N. Lundström.

**Project administration:** Evelina Thunell, Mats J. Olsson, Nina Greilert.

**Resources:** Mats J. Olsson.

**Supervision:** Johan N. Lundström.

**Validation:** Jonas K. Olofsson.

**Visualization:** Arnaud Tognetti, Anja L. Winter, Evelina Thunell, Johan N. Lundström.

**Writing – original draft:** Johannes Frasnelli, Arnaud Tognetti, Sebastian Havervall, Charlotte Thålin, Johan N. Lundström.

**Writing – review & editing:** Johannes Frasnelli, Arnaud Tognetti, Anja L. Winter, Evelina Thunell, Mats J. Olsson, Nina Greilert, Jonas K. Olofsson, Sebastian Havervall, Charlotte Thålin, Johan N. Lundström.

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
