## [Decision Letter · Decision Letter 0]

21 Feb 2024

PONE-D-23-42124High prevalence of long-term olfactory disorders in healthcare workers after COVID-19: a case-control studyPLOS ONE

Dear Dr. Lundstrom,

Thank you for submitting your manuscript to PLOS ONE. After careful consideration, we feel that it has merit but does not fully meet PLOS ONE’s publication criteria as it currently stands. Therefore, we invite you to submit a revised version of the manuscript that addresses the points raised during the review process.

We look forward to receiving your revised manuscript.

Kind regards,

Rong-San Jiang

Academic Editor

PLOS ONE

Journal Requirements:

'Funding was provided by grants awarded to JNL and CT from the Knut and Alice Wallenberg Foundation (KAW 2018.0152 and KAW 2020.0182, respectively), the Swedish Research Council (2021-06527) and a donation from Stiftelsen Bygg-Göta för Vetenskaplig forskning to JNL, as well as the Swedish Heart Lung Foundation to CT. JF is supported by the Fonds de Recherche du Québec – Santé, the Natural Sciences and Engineering Council of Canada (RGPIN-2022-04813) and the Canadian Institutes of Health Research (PJT 173514). No funder took any part in planing, execution, analyses, or reporting of this research."

Reviewers' comments:

Reviewer's Responses to Questions

**Comments to the Author**

1. Is the manuscript technically sound, and do the data support the conclusions?

Reviewer #1: Yes

Reviewer #2: Partly

2. Has the statistical analysis been performed appropriately and rigorously? 

Reviewer #1: Yes

Reviewer #2: Yes

3. Have the authors made all data underlying the findings in their manuscript fully available?

Reviewer #1: Yes

Reviewer #2: Yes

4. Is the manuscript presented in an intelligible fashion and written in standard English?

Reviewer #1: Yes

Reviewer #2: Yes

5. Review Comments to the Author

Reviewer #1: This study explores the persistence of olfactory dysfunctions in healthcare workers who contracted COVID-19 during the initial wave of the pandemic. The researchers aimed to establish the adjusted prevalence of COVID-19-related olfactory dysfunction by comparing results with a control group of SARS-CoV-2 IgG naïve individuals.

Generalizability: The study predominantly focuses on healthcare workers from the first wave of the pandemic, potentially limiting generalizability to other populations with different demographics and health characteristics.

A critical observation is that the most notable results pertain to the sniffin' sticks battery, specifically the TDI, but without presenting evaluations of discrimination, threshold, and identification subtests. It is advisable to include these individual subtest results, identify which is most impacted, and compare/discuss these outcomes with other studies on the general population to discern similarities and differences with healthcare workers. Consider updating the figures accordingly.

Subjective Evaluation: The choice of the VAS scale for subjective evaluation raises questions. Existing literature indicates poor correlation between subjective measures of olfactory dysfunction and psychophysical testing. Preferable alternatives, such as the QOD questionnaire, which also assesses parosmia, could be considered. Given the abundance of articles reporting subjective evaluations in COVID-19 patients, especially during the first wave, the results of the VAS scale could be omitted.

Additionally, the patient and control groups exhibit heterogeneity in numbers.

Minor Edits:

In the Introduction (Lines 76-87), consider condensing this paragraph, as it contains widely known concepts for the audience.

Reviewer #2: The study is interesting and well conducted.

In the introduction I suggest to add the following references :

At page 2 line 58 .Bianco, Maria Rita, et al. "Alteration of smell and taste in asymptomatic and symptomatic COVID-19 patients in Sicily, Italy." Ear, Nose & Throat Journal 100.2_suppl (2021): 182S-185S.

At page 2 line 62. Bianco, M. R., et al. "Evaluation of olfactory dysfunction persistence after COVID-19: a prospective study." European Review for Medical & Pharmacological Sciences 26.3 (2022).

The authors should indicate the TDI score for the various degrees of olfactory dysfunction.

It would be useful to report the correlation between the duration of olfactory dysfunction and the clinical-demographic characteristics of the patients.

In the discussion the authors can add some considerations about the study by Bianco M.R ("Evaluation of olfactory dysfunction persistence after COVID-19: a prospective study"). In this study the authors demonstrated that most patients recovered their sense of smell within the first two months after recovery while a portion (22.2%) still experienced olfactory alterations 4-6 months after SARS-CoV-2 infection. Patients who had not recovered their sense of smell had a significantly longer period of SARS-CoV-2 positivity compared to patients that fully recovered (36.07±7.78 days vs. 29±7.89 days; p= 0.04).

Therefore the recovery time of the sense of smell seems to be related to the duration of the disease; patients who had not recovered their sense of smell had a significantly longer period of virus positivity compared with patients that had recovered their sense of smell .This study for the first time show that recovery of olfactory function depends on the duration of the disease and that the loss of smell can persist even 6 months after healing.

6. PLOS authors have the option to publish the peer review history of their article (what does this mean?). If published, this will include your full peer review and any attached files.

Reviewer #1: No

Reviewer #2: No

---

## [Author Response · Author response to Decision Letter 0]

10 Apr 2024

Journal Requirements:

We have amended files and manuscript in-line with PLOS ONE formatting.

'Funding was provided by grants awarded to JNL and CT from the Knut and Alice Wallenberg Foundation (KAW 2018.0152 and KAW 2020.0182, respectively), the Swedish Research Council (2021-06527) and a donation from Stiftelsen Bygg-Göta för Vetenskaplig forskning to JNL, as well as the Swedish Heart Lung Foundation to CT. JF is supported by the Fonds de Recherche du Québec – Santé, the Natural Sciences and Engineering Council of Canada (RGPIN-2022-04813) and the Canadian Institutes of Health Research (PJT 173514). No funder took any part in planing, execution, analyses, or reporting of this research."

This financial disclosure is now mentioned in the manuscript. 

The funders had no role in study design, data collection and analysis, decision to publish, or preparation of the manuscript. This sentence is mentioned both in the manuscript (lines 335-336) and in the cover letter.

Reviewer #1

1 - Generalizability: The study predominantly focuses on healthcare workers from the first wave of the pandemic, potentially limiting generalizability to other populations with different demographics and health characteristics.

The notion of generalization is a question that we are mindful of and throughout the manuscript, we repeatedly reference the sample as a group of healthcare workers. It can be argued that one can probably not overemphasize this issue and we have therefore added additional pointers. For example, in our final conclusions (lines 314-319), we are now writing (yellow marking to highlight):

“We show that COVID-19 nearly doubles the already large prevalence of quantitative olfactory dysfunction to approximately 37% in a sample of healthcare workers. Furthermore, about half of the COVID-19 survivors exhibit qualitative olfactory dysfunction. Finally, nearly two thirds (65%) of COVID-19 survivors in this group of health care workers exhibit olfactory dysfunction of some form 15 months and 24% reported parosmia 2.6 years after infection. Given the length of time, it is possible that these olfactory problems are not reversable in a plurality of patients.”

In our conclusions, we are now twice highlighting that we are only referring to health care workers. 

However, one should also note that olfactory dysfunction in our control more or less exactly match prevalence numbers in the only two previously published prevalence studies using Swedish samples. This close match suggest that our group of health care workers are not all that different from the general population in respect of olfactory functions.

2 - A critical observation is that the most notable results pertain to the sniffin' sticks battery, specifically the TDI, but without presenting evaluations of discrimination, threshold, and identification subtests. It is advisable to include these individual subtest results, identify which is most impacted, and compare/discuss these outcomes with other studies on the general population to discern similarities and differences with healthcare workers. Consider updating the figures accordingly.

This is an insightful comment that we thank the reviewer for. In response, we have analyzed group differences for the individual subtests and discovered that the main driver of differences is mediated by ability to discriminate and identify odors, suggesting that the higher rate of parosmia in the COVID+ group is a leading contributor to differences. We have incorporated this new information about the individual subtest results into the revised version of the manuscript (lines 172 to 177) and updated Fig 1 to illustrate these interesting results. In addition, we are discussing this new finding in the discussion. We now state in the Result section:

“Interestingly, upon separate analysis of the TDI subscales, only discrimination (W = 1293; N = 139, p = 0.0008; Figure 1C) and identification (W = 1213.5; N = 139, p = 0.0002; Figure 1D) subscores were significantly lower in the COVID+ group (D = 11.66 ± 2.5, I = 12.11 ± 2.5) compared to the COVID- group (D = 13.19 ± 1.9, I = 13.68 ± 1.3). However, no statistically significant difference (W = 2107; N = 139, p = 0.65; Figure 1B) was observed for the threshold subscore between the two groups (COVID+ = 7.10 ± 2.4, COVID- = 7.11 ± 2.0).”

Figure 1. Quantitative olfactory dysfunctions for COVID-19 positive (COVID+, N = 98) and COVID-19 naïve (COVID-, N = 41) groups. (A) Mean TDI scores for COVID+ and COVID- groups showed as bars with error bars denoting standard deviation (SD) as well as values for each participant indicated by circles. Dashed lines indicate cut-off scores for the clinical diagnoses Hyposmia (threshold score: 30.75) and Anosmia (threshold score: 16.25). Higher TDI scores (possible range 1-48) indicate better olfactory performance. TDI subscale scores for the COVID+ and COVID- groups are presented with error bars denoting standard deviation (SD) for (B) Threshold, (C) Discrimination, and (D) Identification scores separately.

Additionally, we are stating in the discussion (lines 263-267):

“Notably, while there was a significant difference in overall olfactory functions between the two groups, as operationalized by the TDI scores, this difference was restricted to performance on the odor quality discrimination and odor identification subtests, but not in odor detection threshold This finding suggests that a potential underlying driver of the differences is the large difference in parosmia that might distort the odor quality of the odors included in the two-odor subtest..”

3 - Subjective Evaluation: The choice of the VAS scale for subjective evaluation raises questions. Existing literature indicates poor correlation between subjective measures of olfactory dysfunction and psychophysical testing. Preferable alternatives, such as the QOD questionnaire, which also assesses parosmia, could be considered. Given the abundance of articles reporting subjective evaluations in COVID-19 patients, especially during the first wave, the results of the VAS scale could be omitted.

We agree with the reviewer that subjective measure of olfactory function is a poor substitute for psychophysical testing. Despite the plethora of studies showing the poor correlation between objective and subjective measurements of olfactory functions, the widespread ignorance of this fact and often sole dependence on subjective ratings of olfactory function is stunning. However, we are explicitly mentioning this fact in several places throughout the introduction. For example, we are stating (lines 55-56): 

“First, most of the estimations are based on self-reported olfactory dysfunction which is notoriously unreliable [12,13].”

That said, we argue that it has a value to show both objective and subjective assessments. After all, what matters most for many individuals are their subjective experience and showing the discrepancy between the two is educational. To further highlight the subpar use of subjective measurements to assess objective olfactory functions, we are now also stating in the introduction (lines 67-68):

“While self-evaluation questionnaires are commonly used to assess quantitative dysfunction, their precision warrants caution in interpretation.”

We hope that the reviewer agrees with us about the value to, once again, highlight the discrepancy between subjective and objective outcomes of olfactory functions.

4 - Additionally, the patient and control groups exhibit heterogeneity in numbers.

We agree that one limitation of our study is the heterogeneity in numbers of our two groups, which we also have prominently mentioned as a limitation in the discussion. However, we mainly performed non-parametric statistical group comparisons using a two-sided Wilcoxon rank sum test with continuity correction that does not assume equal sample sizes and is thus robust against differences in sample size. Still, the relatively limited samples sizes and uneven sample sizes between the two groups remain some limitations that are highlighted in the discussion section. We write lines 302 to 306:

“That said, the strengths mentioned come from the fact that we sampled from a smaller group of closely monitored individuals with a profession that is unfortunately highly taxed for time during a pandemic. This meant that obtaining data from a large and diverse sample was not possible. In addition to the relatively small sample sizes, the COVID+ group had nearly double the participants compared to the COVID- groups, which we aim to statistically control for by using methods such as the Wilcoxon rank sum test. These tests do not assume equal sample sizes and are therefore robust against differences in sample size.” 

5 - Minor Edits:

In the Introduction (Lines 76-87), consider condensing this paragraph, as it contains widely known concepts for the audience.

This paragraph has been shortened. 

Reviewer #2

The study is interesting and well conducted.

Thank you for your positive comment.

1 - In the introduction I suggest to add the following references :

At page 2 line 58 .Bianco, Maria Rita, et al. "Alteration of smell and taste in asymptomatic and symptomatic COVID-19 patients in Sicily, Italy." Ear, Nose & Throat Journal 100.2_suppl (2021): 182S-185S.

At page 2 line 62. Bianco, M. R., et al. "Evaluation of olfactory dysfunction persistence after COVID-19: a prospective study." European Review for Medical & Pharmacological Sciences 26.3 (2022).

Thank you for these relevant references, which have been added to the manuscript. 

2 - The authors should indicate the TDI score for the various degrees of olfactory dysfunction. 

The TDI scores for anosmic, hyposmic, and normosmic individuals from both groups (COVID+ and Control) have been added to the manuscript (lines 180 to 184): 

“A total of 37% of the COVID+ group suffered quantitative olfactory dysfunction, with 4 individuals exhibiting anosmia (TDI score = 14.88 ± 2.0) and 32 individuals exhibiting hyposmia (TDI score = 26.11 ± 3.9). In the COVID- group, 20% (N = 8) showed quantitative olfactory dysfunction, all of whom had hyposmia (TDI score = 29.56 ± 0.8). The average TDI scores for individuals with normosmia were 34.38 (SD = 2.6) and 35.06 (SD = 2.8) for the COVID+ and the COVID- groups, respectively.”

3 - It would be useful to report the correlation between the duration of olfactory dysfunction and the clinical-demographic characteristics of the patients.

While we acknowledge the reviewer's remark, it is essential to note that our knowledge regarding our groups is quite restricted. Our study enlisted hospital staff as participants, not patients currently undergoing hospitalization. Unfortunately, to limit the invasive of their privacy, we lack access to their medical records or comprehensive clinical data, with the sole demographic information gathered being age and sex. Nevertheless, to comply with the request, we assessed the impact of the duration since COVID-19 infection and age on quantitative olfactory dysfunction, we conducted Spearman correlation tests between the TDI score and both the number of days since infection and participants' age. The results of these tests are detailed in the manuscript (lines 184 to 191), and corresponding figures are provided in the supplementary material:

“Last, we investigated whether the degree of quantitative olfactory dysfunction was influenced by the time since COVID-19 infection and individuals' age. A Spearman correlation test indicated that the TDI score of individuals from the COVID+ group was not significantly affected by the number of days since infection (N = 71, r = -.1, p = .43; see Supplementary Figure S1A). However, TDI significantly declined with age in the COVID+ group (Spearman correlation test, r = -.28, p = .005, Supplementary Figure S1B). It's worth noting that in the COVID- group, a similar negative relationship between TDI and age was observed, although it did not reach statistical significance (r = -.27, p = .09, Supplementary Figure S1B).”

Supplementary Figure S1. Scatterplot illustrating the association between TDI score and (A) days since infection or (B) age of individuals who had had COVID-19 (COVID+, in orange) and COVID-19 naïve individuals (COVID-, in blue). Spearman correlations showed that TDI score was not significantly affected by the number of days since infection (N = 71, r = -0.1, p = .43), but significantly declined with age in the COVID+ group (r = -0.28, p = .005) but not in the COVID- group (r = -0.27, p = .09).

4 - In the discussion the authors can add some considerations about the study by Bianco M.R ("Evaluation of olfactory dysfunction persistence after COVID-19: a prospective study"). In this study the authors demonstrated that most patients recovered their sense of smell within the first two months after recovery while a portion (22.2%) still experienced olfactory alterations 4-6 months after SARS-CoV-2 infection. Patients who had not recovered their sense of smell had a significantly longer period of SARS-CoV-2 positivity compared to patients that fully recovered (36.07±7.78 days vs. 29±7.89 days; p= 0.04). Therefore the recovery time of the sense of smell seems to be related to the duration of the disease; patients who had not recovered their sense of smell had a significantly longer period of virus positivity compared with patients that had recovered their sense of smell .This study for the first time show that recovery of olfactory function depends on the duration of the disease and that the loss of smell can persist even 6 months after healing.

We thank the reviewer for bringing this study to our attention and we are now referring to it in the discussion.

---

## [Decision Letter · Decision Letter 1]

17 Jun 2024

High prevalence of long-term olfactory disorders in healthcare workers after COVID-19: a case-control study

PONE-D-23-42124R1

Dear Dr. Lundstrom,

We’re pleased to inform you that your manuscript has been judged scientifically suitable for publication and will be formally accepted for publication once it meets all outstanding technical requirements.

Kind regards,

Rong-San Jiang

Academic Editor

PLOS ONE

Additional Editor Comments (optional):

Reviewers' comments:

Reviewer's Responses to Questions

**Comments to the Author**

1. If the authors have adequately addressed your comments raised in a previous round of review and you feel that this manuscript is now acceptable for publication, you may indicate that here to bypass the “Comments to the Author” section, enter your conflict of interest statement in the “Confidential to Editor” section, and submit your "Accept" recommendation.

Reviewer #1: All comments have been addressed

Reviewer #2: All comments have been addressed

2. Is the manuscript technically sound, and do the data support the conclusions?

Reviewer #1: Yes

Reviewer #2: Yes

3. Has the statistical analysis been performed appropriately and rigorously? 

Reviewer #1: Yes

Reviewer #2: Yes

4. Have the authors made all data underlying the findings in their manuscript fully available?

Reviewer #1: Yes

Reviewer #2: Yes

5. Is the manuscript presented in an intelligible fashion and written in standard English?

Reviewer #1: Yes

Reviewer #2: Yes

6. Review Comments to the Author

Reviewer #1: After the first revision, the manuscript has improved. The authors included all my suggestions and all my concerns have been addressed. Accept as it is.

Reviewer #2: I am satisfied of the reviewers made by the authors to the manuscript and I have no more questions to do.

7. PLOS authors have the option to publish the peer review history of their article (what does this mean?). If published, this will include your full peer review and any attached files.

Reviewer #1: No

Reviewer #2: No

---

## [Editor Report · Acceptance letter]

21 Jun 2024

PONE-D-23-42124R1 

PLOS ONE

Dear Dr. Lundström, 

I'm pleased to inform you that your manuscript has been deemed suitable for publication in PLOS ONE. Congratulations! Your manuscript is now being handed over to our production team.

Kind regards, 

on behalf of

Dr. Rong-San Jiang 

Academic Editor

PLOS ONE